



# Characterisation of Intra-hourly Wind Power Ramps at the Wind Farm Scale and Associated Processes

Mathieu Pichault[1], Claire Vincent[2], Grant Skidmore[1], and Jason Monty[1]

[1]Department of Mechanical Engineering, The University of Melbourne, Victoria 3010 Australia
[2]School of Earth Sciences, The University of Melbourne, Victoria 3010 Australia

**Correspondence:** Mathieu Pichault (mpichault@student.unimelb.edu.au)

**Abstract.** One of the main factors contributing to wind power forecast inaccuracies is the occurrence of large changes in wind power output over a short amount of time, also called 'ramp events'. In this paper, we assess the behaviour and causality of 1183 ramp events at a large wind farm site located in Victoria (southeast Australia). We address the relative importance of primary engineering and meteorological processes inducing ramps through an automatic ramp categorisation scheme. Ramp features such as ramp amplitude, shape, diurnal cycle and seasonality, are further discussed and several case studies are presented. It is shown that ramps at the study site are mostly associated with frontal activity (46%) and convective processes (29%), and that wind power fluctuations tend to plateau before and after the ramps. The research further demonstrates the wide range of temporal scales and behaviours inherent to intra-hourly wind power ramps at the wind farm scale.

## 1 Introduction

Environmental protection and sustainability have become the main incentives to integrate more green energy sources into electrical systems. Numerous countries are currently moving towards greener energy production sources to achieve the Paris Agreement's goal to keep global warming below +2° by 2100 (UNFCCC, 2015). Since the early 2000s, wind energy has gained significant traction and is currently the fastest-growing mode of electricity production across the globe (EIA, 2019), with up to 51.3 GW of wind power capacity installed worldwide in the year 2018 alone (GWEC, 2019). In emerging markets such as Australia, Canada and the United States, newly built wind farms are installed in large blocks, often exceeding 400 MW (Kariniotakis, 2017). With an ever-growing wind penetration in the grid, electricity networks are increasingly subject to fluctuations in power production. These fluctuations are called 'ramp events', referring to the sudden variations in wind power generation over a short period of time. Motivated by the need to enhance management of such events as well as optimising integration and control of wind farms, there is currently a great demand incentive to develop accurate and timely short-term (intra-hourly) ramp forecasts (Zhang et al., 2017; Cui et al., 2015; Gallego et al., 2015a).



Sharp increases ('ramp-up') or decreases ('ramp-down') in wind power generation over a short period of time give rise to both financial and physical impacts. First, wind power ramps are a risk to electric system stability and their mismanagement can have dramatic consequences, such as power outages (Tayal, 2017; Trombe et al., 2012). These can be particularly detrimental
to electrical networks located in areas with a low degree of inter-connectivity (i.e. remote regions or islands), where significant power variations are not easily balanced (van Kooten, 2010; Treinish and Treinish, 2013). Both ramp-ups and ramp-downs can exhibit diverse levels of severity (i.e. likelihood to cause disturbances) according to the time and geographic scale over which the ramp occurs (Zhang et al., 2014). However, ramp-downs are generally considered more likely to impact grid system stability due to the limited availability of reserve power (Zhang et al., 2017; Jørgensen and Mohrlen, 2008). Additionally, wind
farms are often curtailed during ramp-ups as electricity surplus cannot be dispatched, which represents loss of potential profits for wind farm owners. In many cases, wind farm owners also have to cover additional costs when they are unable to meet specific loads and quotas.

Improved ramp prediction can help mitigate the issues listed above. However, wind power ramps are particularly challenging to predict. This is partly due to the wide variety of time scales over which they occur, ranging from a few minutes up to several
hours (Worsnop et al., 2018). At the wind farm scale, numerical weather prediction models struggle with forecasting wind power fluctuations occurring within an hour and often fail to predict accurately the timing and the amplitude of the ramps (Zack et al., 2011; Magerman, 2014). In practice, the vast majority of operational short-term wind forecasts rely primarily on variations of the persistence method (or 'naive predictor') (Wurth et al., 2018), which assumes that there will be no variation between the current conditions and the conditions at the time of the forecast. Persistence forecasts inherently tend to perform
poorly during ramp events.

Wind power ramps are usually characterised by their magnitude $\Delta P$, duration $\Delta t$, rate $\Delta P/\Delta t$, timing $t_0$ (central time or starting time of the event) and their gradient direction (ramp-up or ramp-down) (Sherry and Rival, 2015; Lange et al., 2010; Ferreira et al., 2010). However, defining a ramp event is a non-trivial task. In fact, there is currently no commonly agreed upon definition for a wind power ramp (Gallego et al., 2015a; Mishra et al., 2017) as its interpretation can vary substantially between
applications (Wurth et al., 2019; Cutler et al., 2007; Bradford et al., 2011; Greaves et al., 2009). In addition, some operators may need to to evaluate the likelihood of ramps occuring based on various definitions simultaneously (Bianco et al., 2016). Many wind power ramp studies employ a binary, threshold-crossing identification system. However, these binary identification systems are limited by the high sensitivity of the definition to the adopted threshold. Furthermore, it implies all ramps are identical and does not provide further insights on their severity. To alleviate these shortcomings, the so-called 'ramp functions'
have been introduced, which provide an estimation of the ramp intensity at each time step. Gallego et al. (2013, 2014) first introduced a ramp function based on a continuous wavelet transform (the 'Haar' Wavelet) of a wind power time series. More recently, a continuous wavelet transform (CWT) based on a Gaussian wavelet was used by Hannesdóttir and Kelly (2019) and Martínez-Arellano et al. (2014) proposed a ramp function based on a fuzzy logic approach to characterise ramps for the day-ahead market.

A precursor to successful ramp predictions is a sound understanding of the conditions under which ramps occur (Couto et al., 2015). Identifying the temporal and spatial scales pertaining to ramps also provides valuable insights on the limits of





numerical weather prediction models and associated uncertainties (Gallego et al., 2015a). Nonetheless, ramping behaviour analysis is a relatively new research field and very little is known about the main processes inducing ramps (Mishra et al., 2017). In Cutler (2009), approximately 40% of the ramps observed at three Australian wind farms were associated with frontal
systems, while neighbouring high and low-pressure systems and troughs accounted for 35% of the ramps. In Jørgensen and Mohrlen (2008) and Sherry and Rival (2015), the authors observed a strong correlation between ramp events and Chinook (Föhn) winds days, emphasising the importance of local meteorological events in forming ramps. Deppe et al. (2012) found the presence of low-level jets was the primary driver of ramps at a site located in Pomeroy (IA, USA). Based on a more extensive data set, Walton (2012) suggested most of the ramps at the Pomeroy site were instead associated with thunderstorms
and steep pressure gradients. Other studies in Central Europe have shown that most critical ramps arise from extreme weather events such as cyclones (Steiner et al., 2017; Madalena Lacerda et al., 2017). These findings suggest a relatively high degree of correlation between ramping behaviour and large-scale atmospheric circulation processes, emphasising the great potential to use synoptic weather typing and operational decision tools to support power system with a high degree of wind penetration. Although discussed in multiple studies (Deppe et al., 2012; Ferreira et al., 2012; Freedman et al., 2008; Kamath, 2010; Sherry
and Rival, 2015), there is no consensus in the literature on seasonal and diurnal ramp patterns, underlining the influence of local features on ramping behaviour. In summary, we see that the expected main drivers of ramps can vary significantly according to geographic location and that site-specific conditions such as terrain roughness, orography, air-sea-land interactions play a critical role in inducing ramps at the wind farm scale.

As pointed out by Cutler et al. (2007), Gallego et al. (2013, 2015b), and Mishra et al. (2017), robust ramp classifications are
still currently needed owing to the emerging nature of the subject. Review of the literature revealed that while studies assessing the causality of wind power ramps exist, these focus mostly on a limited number of critical events rather than more frequent fluctuations. The lack of clear identification criteria prevents the implementation of automatic classification schemes and hence precludes tracing the causality of more common (less severe) ramps. Hence, there are both scientific and practical interests to develop automatic schemes for (intra-hourly) ramp categorisation. This study aims at characterising intra-hourly wind power
ramps and their underlying processes at the wind farm scale through such an approach. The paper is organised as follows. The methodology to detect ramps and extract relevant features, as well as to categorise ramps according to their underlying processes is established in Section 2. Section 3.1 provides details on the main ramp features at the study site. Section 3.2 addresses the underlying meteorological and engineering causes of ramps, and ramp shapes are discussed in Section 3.3. For illustrative purposes, characteristic case studies are presented in Section 3.4. Finally, conclusions and a discussion of future
works are presented in Section 4.

## 2   Methodology

### 2.1   Data

Data were collected at the Mount Mercer wind farm ('the site') in Western Victoria (southeastern Australia). The site comprises a 2650 ha area of moderately complex topography. The prevailing wind direction at the site is north-northwest, with occasional





westerly and southeasterly winds. Sixty-four (64) identical doubly-fed induction generator wind turbines manufactured by
Senvion (model $MM92$) of 2.05 MW nominal rated capacity are installed onsite, corresponding to a total installed capacity of
131.2 MW. The power curve characteristic of wind turbines is presented in Figure 1. The wind turbines are expected to reach
their rated capacity for wind speed above 11 ms$^{-1}$. The cut-in speed of the wind turbines is 3 ms$^{-1}$.

All data analysed as part of this study were collected between 01/10/2016 and 01/03/2019. Power data consist of one minute
averaged total power generation of the wind park. Outliers and periods of abnormal operation were filtered out of the power
generation time series. Wind data collected at the site as part of this assessment include one second-resolution (1 Hz) wind
speed and wind direction measurement. Wind data originate from two 80 metre high meteorological towers ('met masts'),
$MM1$ and $MM2$, each of them comprising two cup anemometers installed at 78 and 80 m above ground level (AGL) and
two wind vanes installed at 35 and 76 m AGL. $MM1$ and $MM2$ are located in the northwest and southeast corner of the site,
respectively. Figure 2 shows the layout of the wind farm along with the location of the turbines and met masts. Additional
precipitation data were collected from the Australian Bureau of Meteorology Sheoaks weather station (Latitude: -37.910000,
Longitude: 144.130000), located approximately 25 km southeast of the site.

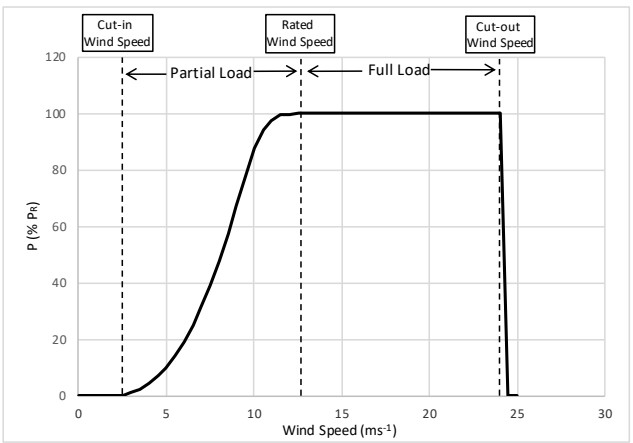

**Figure 1.** Manufacturer's power curve of onsite turbines (Senvion $MM92$).

## 2.2   Ramp Function

Ramps typically correspond to localised sudden changes in a wind power time series. It is possible to characterise ramps based
on the notion that ramps occur when a specific large gradient is maintained during successive time steps in a wind power time
series through so-called 'ramp functions'. The main focus of ramp functions is to provide an estimation of the intensity of the
ramp at each time step of a wind power time series.

Wavelet analysis has shown to be a powerful tool to study variations in local averages (Percival and Walden, 2000). CWT
have been used in the wind energy space to characterize wind power ramps (Gallego et al., 2013, 2014) and wind speed ramps
(Hannesdóttir and Kelly, 2019). The continuous wavelet transform (CWT) enables decomposition of a time-amplitude signal



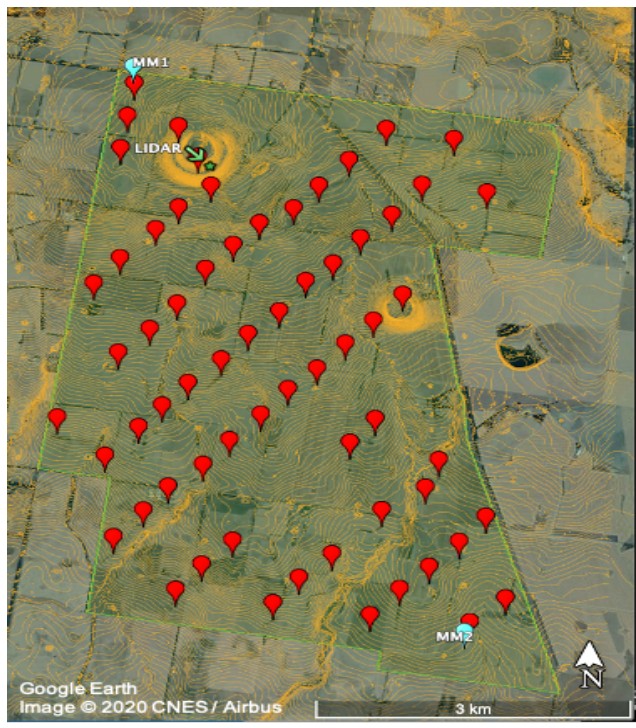

**Figure 2.** Mount Mercer wind farm location Plan. The location of the 64 turbines and the 2 met masts are designated by the red and blue markers, respectively.

in the time-frequency domain and hence provides information on the timing $t'$ and the scale $\gamma$ of particular events. Briefly stated, the CWT is obtained by computing the dot product between a signal $p_t$ and a mother wavelet $\Psi(t)$ which has been transformed through shifting and stretching operations:

$$W_p(t',\gamma) = \frac{1}{\gamma} . \int_{-\infty}^{\infty} p_t . \Psi\left(\frac{t-t'}{\gamma}\right) \tag{1}$$

where $W_p$ are the wavelet transform coefficients, which are functions of the scale dilatation $\gamma$ and time shift $t'$.

In the context of this study, a ramp function following the procedure outlined by Gallego et al. (2013) was implemented. For sake of completeness, the methods and equations as per Gallego et al. (2013) are presented in the remainder of this section. The approach is based on the CWT of the 'Haar' Wavelet to provide an estimation of the ramp intensity at each time step. Amongst the numerous existing wavelet forms, the Haar wavelet was chosen for the adopted methodology because of its capacity to
quantify the gradient of a signal at various time scales (Percival and Walden, 2000). The coefficients resulting from the CWT



based on the Haar wavelet transform of a wind power time-series $p_t$ are denoted $W_p(t,\gamma)$ and expressed by:

$$W_p(t,\gamma) = \begin{cases} \frac{1}{\gamma} \cdot \left( \sum_{i=1}^{i=\gamma/2} p_{t+i-1} - \sum_{i=1}^{i=\gamma/2} p_{t-i} \right) \\ \qquad \text{if } \gamma \text{ is even} \\[2ex] \frac{1}{\gamma} \cdot \left( \sum_{i=1}^{i=(\gamma-1)/2} p_{t+i} - \sum_{i=1}^{i=(\gamma-1)/2} p_{t-i} \right) \\ \qquad \text{if } \gamma \text{ is odd} \end{cases} \tag{2}$$

Note that the coefficients described in Equation 2 are derived from the additive inverse of the conventional Haar wavelet (obtained by changing the sign of the mother wavelet). This was done so as to obtain coefficients whose signs are equal to the sign of the gradient experienced by the time series. The ramp function $R_t$ is then defined as the sum of each WT coefficients calculated for the scale interval $[\gamma_1, \gamma_N]$:

$$R_t = \sum_{\gamma=\gamma_1}^{\gamma_n} W_{t,\gamma} \tag{3}$$

The ramp function (or 'Ramp score') is considered a reflection of the ramp intensity as the contribution to the gradient is assessed for different scales at each time step. The ramp function defined by Equation 3 can be normalised by its maximum absolute value to generate the normalised ramp function $R_{\text{norm } t}$, comprised between 1 (the strongest ramp-up) and $-1$ (the strongest ramp-down).

## 2.3 Ramp Detection and Characterisation

First, a wavelet-based ramp function (Gallego et al., 2013) is computed to quantify the ramp performance (or 'score') at each time step of the time series. The ramp function is computed on the time series using a scale range $[\gamma_1, \gamma_N]$ of $[2, 60]$, therefore primarily targeting ramps occurring over a maximum time window of 60 minutes (intra-hourly ramps). Times $t'$ of the most significant ramps are identified by selecting the 1% of events associated with the strongest absolute ramp intensity (i.e. times associated with highest ramp scores). The maximum wavelet coefficient at the timing of the ramp determines the time scale $\gamma$ (or 'scale') of the ramp. The approach which consists of identifying ramps' timescales based on the maximum wavelet coefficient was first implemented in a study by Hannesdóttir and Kelly (2019), in which ramps in wind speed are studied. The subset of the wind power time series of length $\gamma$ and centred on $t'$ will henceforth be referred to as 'the ramp'.

Finally, two ramp features, namely the ramp amplitude and the rise time, are retrieved. The ramp amplitude, denoted $\Delta P$, is defined as the maximum power variation over the subset length $\gamma$ centred on $t'$. The rise time hereafter refers to the elapsed time between the lowest and highest power level during the ramp.

Figure 3 illustrates the decomposition of a wind power generation time series into its wavelet coefficients. The thickened red line on Figure 3 (a) represents the ramp of scale $\gamma = 20$ minutes and the vertical and horizontal arrows indicate the amplitude (65 MW) and the rise time (11 minutes), respectively. The red dot on Figure 3 (b) displays the highest absolute coefficient value at the timing of the ramp, which corresponds to a scale $\gamma$ of 20 minutes. Note that the longest ramp scale to be identified through this method is 60 minutes since the ramp function is calculated with an upper scale range limit $\gamma_{max}$ of 60.





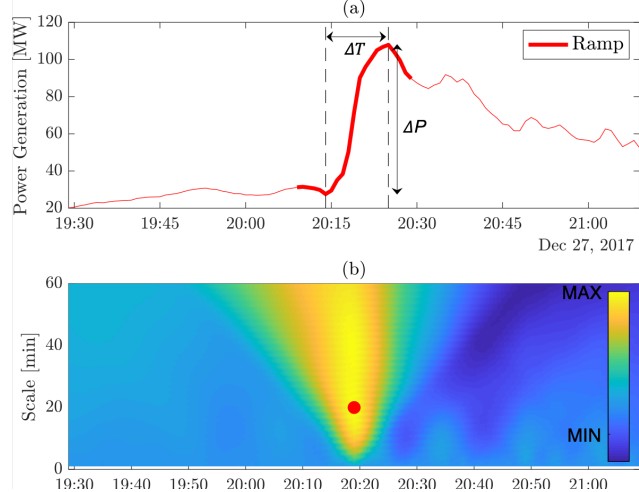

**Figure 3.** (a) power generation time series surrounding ramp rank ID number 5 and (b) coefficients of its continuous wavelet transform based on the 'Haar' wavelet. The ramp timing (central point of the ramp) is on 27th of December 2017 at 20:19. The thickened red line on Figure (a) represents the ramp, whose amplitude $\Delta P$ is 65 MW and rise time $\Delta T$ is 11 minutes. The red dot on figure (b) indicates the maximum coefficient at the timing of the ramp, corresponding to scale $\gamma$ of 20 minutes.

In order to avoid ramp over-identification (i.e. identifying variations of the same event multiple times), ramps associated
with a lower score occurring within the scale range of a more significant ramp are discarded. Based on the methodology above, a total of 1183 ramps are identified in the 29 month period.

## 2.4    Ramp Categorisation

In this section, we present the automatic scheme developed to classify ramps according to their underlying causes. While it is expected that processes associated with ramps will present common structural features, their realisations are, in essence, unique
and might involve a range of factors. Therefore, the goal here is to identify the most-likely ramp driver as opposed to capturing every possible scenario that could have resulted in a ramp. The method is designed to automate ramp classification based on easily accessible data. The data set of ramps to-be-classified exclude ramps occurring during periods of environmental sensor failure, identified by constant readings. The filtering process effectively removed 8 ramps from the original data set. For the sake of clarity, a decision tree used to diagnose ramp driver categories is summarised in Figure 5. Based on the review of the
literature on ramp drivers, we established criteria to classify ramps in six categories. The criteria are as follows (by order of priority):

– **Passage of a Front**: Weather fronts are caused by abrupt changes in air mass and are often associated with strong low-altitude winds, precipitations and a shift in wind direction. Such macro-scale meteorological processes often incur large wind power generation fluctuations (increase caused by the passage of the front and decrease caused by the pre-frontal
lull or post-frontal relaxation). The criterion to identify fronts is adapted from the Melbourne Frontal Tracking System



(Simmonds et al., 2012), initially developed to explore cold front behaviour in the Southern Hemisphere from reanalysis data. The method was selected by the authors amongst numerous objective algorithms due to its ability to identify fronts in the southern hemisphere with remarkable veracity while preserving a straightforward, easily understandable scheme. The front identification scheme is summarised as follows: (1) the sign of the meridional component of the wind ($v$) changes from positive to negative over successive time points $[t, (t + 6h)]$ (i.e. the wind direction shifts from the southwest to the northwest quadrant), and (2) the amplitude change of the meridional wind component is larger than 2 ms$^{-1}$ over the same 6-hour interval. As the method was originally developed for the $ERA\ Interim$ reanalysis (Dee et al., 2011) on a $1.5^o$ latitude-longitude (approx 160 km) grid, the objective function was further adapted to process data from discrete spatial points by applying a 4-hour moving average filter to the wind data time series. The use of such an averaging window is justified by the fact that spatial averaging over a 160 km grid is somewhat equivalent to temporal averaging over approximately 4 hours, assuming an average wind speed during ramps of 11 ms$^{-1}$. Additionally, computing the adapted front detection algorithm using a 4-hour averaging window provided high agreement with front identification through inspection of MSLP charts. In short, the objective scheme for ramps associated with frontal passages is a wind shift from the southwest to the northwest quadrant combined with a change of meridional wind greater than 2 ms$^{-1}$ when comparing wind conditions with a 4-hour moving average 3 hours before and after the ramp timing. The reader interested in implementing a similar automatic front scheme targeting the northern hemisphere is referred to Bitsa et al. (2019).

– **Post Frontal Activity**: To capture the strongly variable wind conditions following the passage of a front, where cold-air outbreaks and cellular convection often dominate the flow fields, ramps occurring within 12 hours after the passage of a cold front but not related to the front itself were labelled as post-frontal driven. The adapted front identification scheme introduced above was applied to the entire wind time series with an hourly time-step, and the most likely timing for each front was selected with a cumulative sum over a 6-hour window. A total of 394 fronts were identified following this methodology, which corresponds to a front occurring approximately 2 percent of the time within the time series.

– **Moist Convection**: The passage of thunderstorms or showers can also cause ramps. A ramp is deemed to be associated with moist convection processes (*e.g.* convective outflow or gust fronts) if at least 0.2 mm of precipitation was detected at the Sheaoks meteorological weather station neighbouring the site during the ramp or up to one hour before the ramp.

– **High Turbulence**: Ramp-up and ramp-down may be due to turbulent flow. Since temperature measurements collected at different heights, required for stability classification based on the Richardson number, were unavailable at the time of the investigation, we adopted a simplified stability classification based on turbulence intensity (TI) at hub height. We computed the horizontal turbulence intensity from the onsite cup anemometers $TI_{Ucup} = \frac{\sigma_U}{\overline{U}}$, where $\sigma_U\ [ms^{-1}]$ and $\overline{U}$ $[ms^{-1}]$ are the average standard deviation of the horizontal wind speed and the average horizontal wind speed over a 10 min period, respectively. For reference, examples of TI thresholds for stability conditions classification encountered in the literature are presented in Table 1. The distribution of TI observed at the site during the assessment period is shown in Figure 4. We relate ramps to highly turbulent conditions when the associated TI during the ramp scale preceding the



ramp exceeds 14%. We chose the threshold suggested by Wharton and Lundquist (2012) as it was derived explicitly targeting horizontal TI and generally consistent with onsite TI observations. The rationale behind considering TI prior to a ramp is to reflect as much as possible the background conditions in which the ramp took place instead of being subject to the ramp itself.

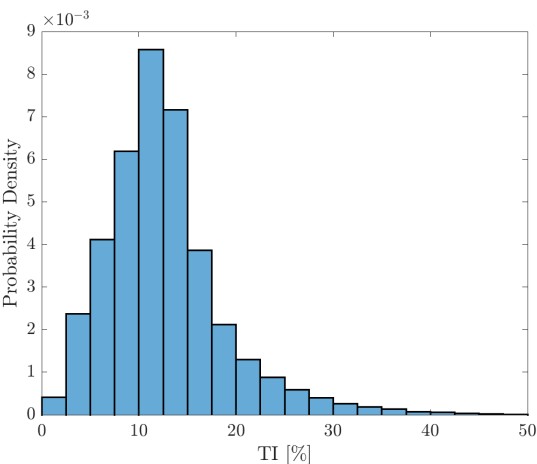

**Figure 4.** Frequency distribution of turbulence intensity at the Mount Mercer wind farm between the 01/10/2016 and the 03/05/2019.

**Table 1.** Review of TI criteria for stability and turbulence regime classification

|  | Wharton and Lundquist (2012) | Hansen et al. (2012) | Rareshide et al. (2009) | Bardal and Sætran (2017) | Belu and Koracin (2012) |
|---|---|---|---|---|---|
| Stable | TI < 9% | TI ≈ 5% | - | - | - |
| Neutral | 9% < TI < 12% | - | - | - | - |
| Convective | 12% < TI < 14% | TI ≈ 7% | - | - | - |
| Strongly Convective | 14% < TI | - | - | - | - |
| Low Turbulence | - | - | 5% < TI < 11% | 0% < TI < 10% | 2% < TI < 8% |
| High turbulence | - | - | 11% < TI < 17% | 10% < TI | 8% < TI < 15% |

– **Stable Boundary Layer Effect**: Stable atmospheric boundary layer conditions can produce ramp-ups due to decoupling
of the boundary layer and onset of low-level jets, or ramp-downs due to onset of strong stratification of the boundary layer with generally light winds. In a similar fashion to the 'high turbulence' category, we related ramps to stable boundary layer effects when the associated TI during the ramp scale preceding the ramp is less than 9%.

– **Shutdown cases**: Small changes in wind speed can have a significant impact on power production when velocities are close to the wind turbines' cut out wind speed (wind speed above which the turbine will automatically shut down for
safety reasons). Hence, ramps were labelled as shutdown situations when the maximum 10 minutes averaged wind speed measured at the site exceeded the cut-out wind speed (24 ms$^{-1}$) during the ramp.



    – **Non-linearity of the power curve**: Although sudden changes in wind speed usually cause significant power variations, minor variations in a moderate wind regime can also lead to substantial wind power fluctuation. This is due to the non-linear relationship between wind speed and power generation, shown in Figure 1. Non-linearity of the power curve is not a driver of its own, but this ramp class aims to complement previously introduced drivers and as such will be considered separately. Ramps are associated with the non-linear wind-to-power conversion processes when the amplitude in 10 minutes-averaged wind speed is less than 3 ms$^{-1}$ and comprised within the 5-10 ms$^{-1}$ window (i.e. the steep portion of the power curve).

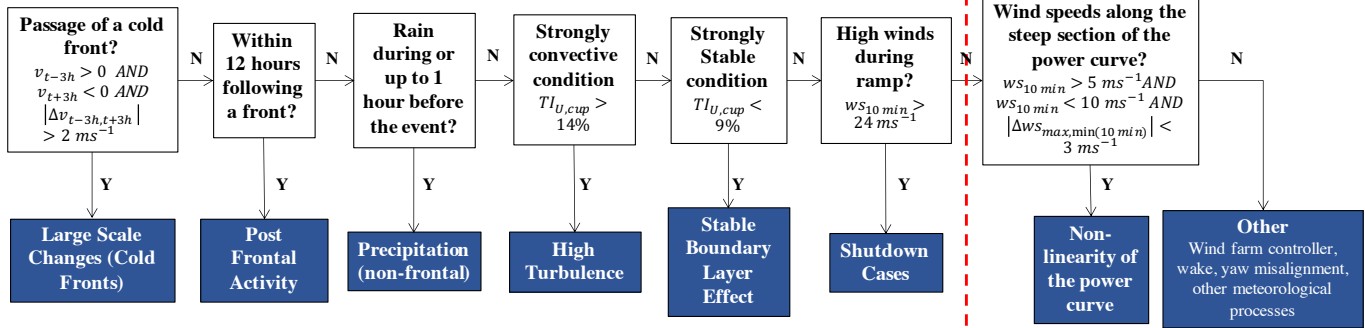

**Figure 5.** Schematic of the ramp associated meteorological and engineering drivers decision tree.

## 2.5 Ramp Shape

As part of the ramp characterisation, the overall shapes of the power fluctuations encompassing ramps were investigated, with a view to answering the question of whether particular ramp-drivers have a characteristic ramp shape. A subset of the power generation time-series is extracted for each ramp identified above. The subset is centred on the ramp timing $t'$ and of duration equal to three rise times ($3\Delta T$). The wind power subset with a duration of three scales and centred on the ramp timing will be hereafter referred to as the 'extended ramp'. To compare the shape of extended ramps, all subsets are then normalised by their corresponding ramp amplitude and rise time. Finally, time-series are clustered in 8 groups using self-organising maps (SOM).

    SOM is a method for clustering data based on similarity using artificial neural networks (Kohonen, 1982). Based on the assumption that power fluctuations can either level-out or follow an inverse trend before and after the ramp, we can logically expect eight categories of shapes. For that reason, the SOM configured as part of this study is a 2x4 layers neural network, hence identifying eight synoptic ramping behaviour classes.

    Finally, we used a re-sampling technique with replacement method, also referred to as 'bootstrapping' (Efron, 1979), to investigate potential interactions between the shape and the driver associated with a ramp. The 95% confidence interval around the mean is derived from a bootstrap method with 1000 re-samples in which ensemble members are assessed against the distribution of independent samples. For a brief description of the implementation of the bootstrapping method, interested readers are referred to the Appendix of this paper.





## 3  Results

### 3.1  Ramp characteristics and behaviour/occurrence

The ramp detection scheme described in Section 2.2 identified a total of 1183 ramps, which accounts for 5.16% of the total time series (in terms of total ramping time). The distributions of the ramp rise-time and amplitude, along with the relationship between them are presented in Figure 6. As shown in Figure 6, the amplitudes and the rise times of the intra-hourly ramps at the site range from $\Delta P \in [29.0 \, , 120.3 \,]$ MW and $\Delta t \in [4 \, , 60 \,]$ min, respectively and most common intra-hourly ramp features at the site consist of a variation of power of 46.0 MW (35% rated capacity) and a rise time of 28 min.

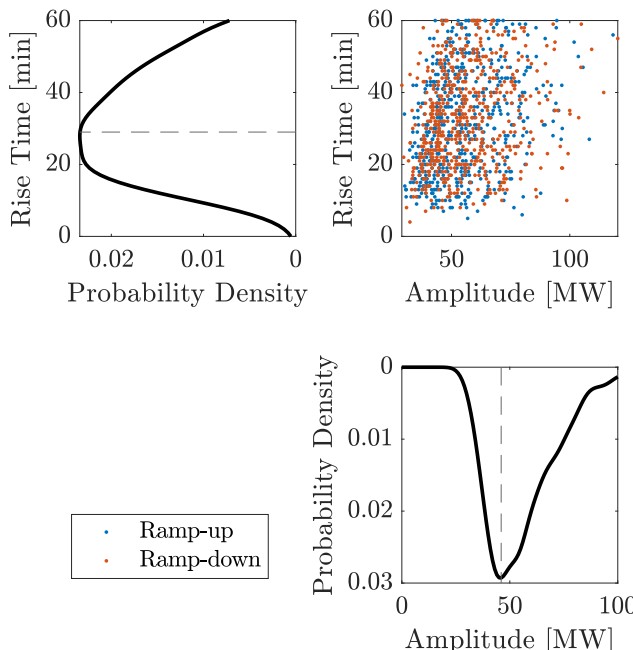

**Figure 6.** Relationship between the ramp amplitude and rise time amongst the ramp data set (top right corner) and associated Kernel probability density distributions (top left and bottom right corner, respectively). The blue and orange dots designates ramp-ups and ramp-downs, respectively. The dashed lines display the modes of the distributions.

Figure 7a displays the hourly distribution of ramp-ups and ramp-downs for the data set. Both upwards and downwards ramps exhibit higher propensity of occurrence during daylight hours, with a moderate peak of upwards ramps at 10 am. On the other hand, downwards ramps tend to occur late afternoon, with a maximum likelihood of occurrence at 15:00. Both upwards and downwards ramps are more common during warmer months, with a noticeable peak in Spring (Figure 7b).

Amongst the 1183  ramps analysed, 590  (49.87 %) were ramp-ups. The equal proportion between ramp-ups and ramp-downs exhibited in the data set could be seen as a contradiction, with other studies (Freedman et al., 2008; Ferreira et al., 2012; Kamath, 2010; Jørgensen and Mohrlen, 2008) suggesting ramp-ups are more frequent as they often result from rapidly moving

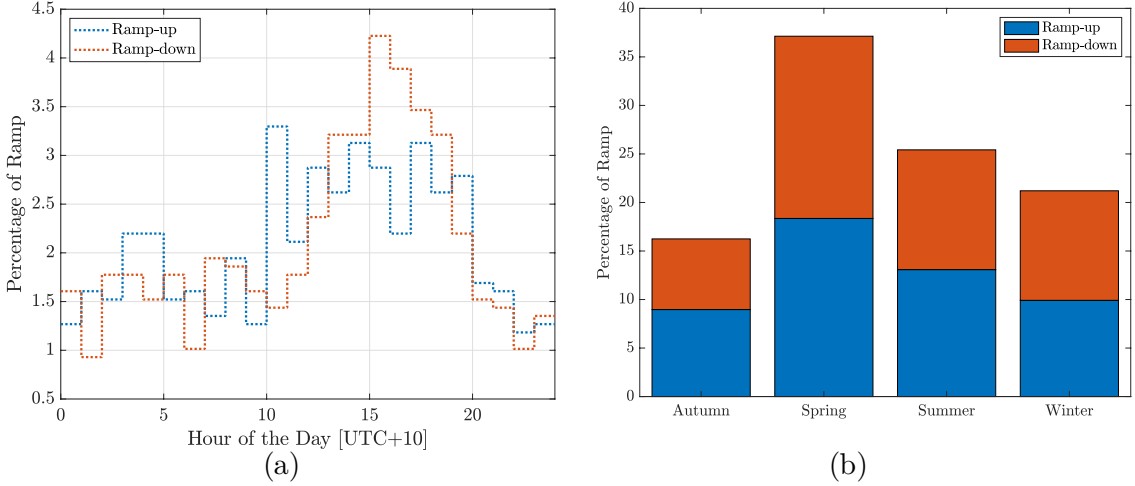

**Figure 7.** (a) hourly ramp distribution and (b) bar plot of the seasonality of the ramps, based on data comprised between 01/01/2017 and 01/01/2019.

transient features causing a sharp increase in wind power followed by a gradual decrease (Freedman et al., 2008; Ferreira et al.,

2012). However, the even ratio between ramp-ups and ramp-downs observed in this study is a direct consequence of the broad temporal coverage of ramps considered within the power generation time series. It is naturally expected that the proportion of ramp-ups vs ramp-downs converges towards one as the proportion of investigated variability increases. This behaviour is depicted in Figure 8 showing the ratio of ramp-ups/ramp-downs as a function of the number of strongest ramps considered in the data set. When considering larger power fluctuations (i.e. ramps associated with higher normalised ramp scores $R_{norm}$),

the ratio ramp-up/ramp-down increases as stronger ramps are more frequently ramp-ups. These findings are thus consistent with previous studies.

### 3.2   Ramp Categorisation

Figure 9 presents the distribution of ramps according to their underlying processes. The proportion of each driver category (following the decision tree on Figure 5) and each criterion considered alone, together with the total number of ramps for each

driver category are provided in Table 2. 46 % of the intra-hourly ramps at the wind farm site are related to frontal activity, with cold fronts and post-frontal conditions accounting for 21% and 25% of the ramps, respectively. Precipitation and high turbulence events accounted for 12% and 17% of the ramps. Stable ABL effects only explained 6% of the ramps and times during which the wind farm production is externally restrained due to high winds are relatively rare, accounting for only 0.6% of the ramps within the data set. Up to 8% of the ramps were associated with small variations in wind speed along the 5-10

265   ms$^{-1}$ section of the power curve. When considered separately, the 'non-linearity of the power curve' criteria was 'satisfied' for approximately 37% of the ramps (Table 2), hence strengthening the importance of accurate wind predictions within this range. Overall, all drivers instigate both upwards and downwards ramps, as per the expected processes described in Section 2.5.

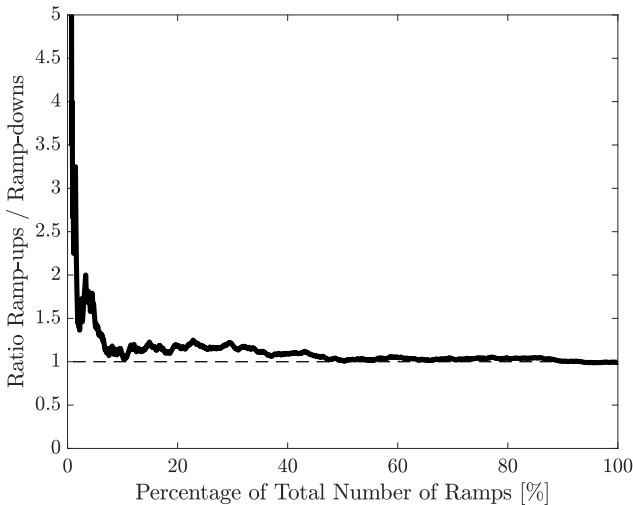

**Figure 8.** Proportion of ramp-up as a function of the ramp score.

As shown in Figure 10a representing the diurnal distribution of cold front ramp-ups against all other ramp-ups, the observed 10 am peak in ramp-ups observed in Figure 7a appears to be closely related to timing of frontal passages. In fact, most 10 am ramp-ups are associated with cold fronts (41%). Likewise, the majority (42%) of downwards ramps observed at 15:00 are associated with post-frontal conditions (explained by the relaxation of wind speed after the passage of a front causing downward ramps). To investigate this further, we consider the hourly distribution of frontal passages during the assessment period based on the adaptation of Simmonds et al. (2012) front identification algorithm discussed above (Figure 10b), where a clear 10 am maximum frequency of incidence is observed. These findings are consistent with the findings of Berson et al. (1957), in which an early afternoon maximum in frontal passage was reported in the region.

**Table 2.** Results of the automatic ramp classification scheme.

|  | Cold Fronts | Post Frontal Activity | Precipitation (non-frontal) | High Turbulence | Stable ABL Effects | Shutdown Cases | Non-linearity of the Power Curve | Other |
|---|---|---|---|---|---|---|---|---|
| Decision Tree [%] | 20.77 | 25.02 | 11.83 | 17.28 | 6.04 | 0.6 | 8.34 | 10.13 |
| Absolute [%] | 20.77 | 25.02 | 28.09 | 37.53 | 9.53 | 0.68 | 36.77 | 10.13 |
| Nbr Ramp-up | 109 | 135 | 64 | 127 | 39 | 3 | 61 | 52 |
| Nbr Ramp-down | 135 | 159 | 75 | 76 | 32 | 4 | 37 | 75 |

After classification, 10% of the ramps could not be directly attributed to one of the categories introduced above. While the proposed approach effectively portrays the prevalence of main ramp drivers, we acknowledge the method cannot capture all possible events inducing ramps. For example, meteorological phenomena such as micro-bursts, gravity waves, intermittent





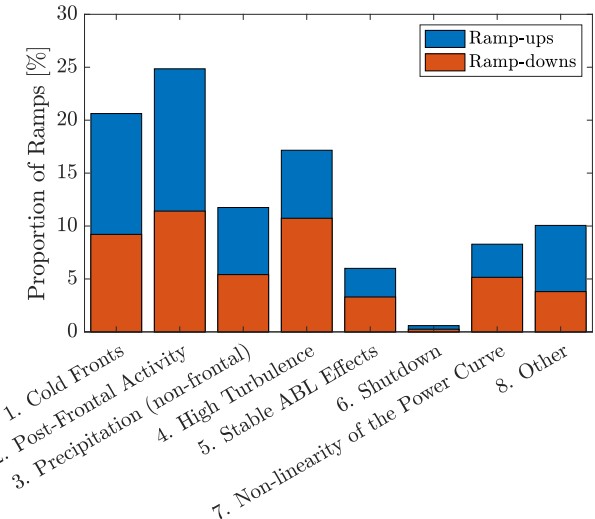

**Figure 9.** Categorisation of intra-hourly wind power ramps between 01/10/2016 and 01/03/2019.

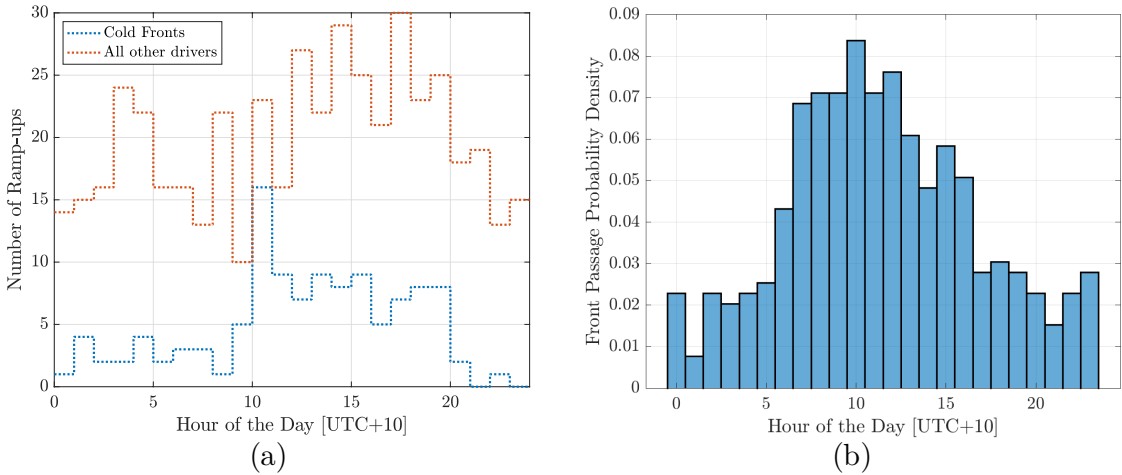

**Figure 10.** (a) hourly distribution of ramp associated with cold fronts vs all other drivers, and (b) hourly front distribution based on the adapted Melbourne Frontal Tracking System between 01/10/2016 and 03/05/2019.

cloud cover and low-level jets were not directly assessed, although they may implicitly appear in some of the categories. In addition, the ramp classification does not consider other mechanical processes such as wake effects and yaw misalignment (their contribution in explaining ramp events is expected to be marginal). Finally, the front detection algorithm could not identify fronts with perfect accuracy. As discussed in Section 3.4, some ramps classified as 'other' are in fact associated with the passage of fronts or troughs.





### 3.3 Extended Ramp Shape

Figure 11 displays the variety of power fluctuation behaviours obtained through self-organising maps with 8 groups (SOM1-8). For clarity, the thick black lines represent the mean extended ramp shape and the grayscale contours display the point density for each SOM group. The dashed lines delineate periods equal to one rise time. The trend of the fluctuations before, during and after the ramp is characterised by $[x,y,z]$, in which x, y and z can exhibit three discrete values: 0 for plateauing trend, -1 for downwards trend and 1 for upwards trend. The relative proportion of ramps within each group is also indicated on Figure 11. The categories resulting from the SOM correspond to the 8 shape classes expected when assuming power generation can either level-out or follow an inverse trend before and after the ramp.

While results from the SOMs indicates a great variability in extended ramp behaviour, specific shape classes exhibit different patterns. In particular, fluctuations commonly tend to plateau before and after the ramp, with such behaviour being observed for 44% of the ramps (22% for both $[0,1,0]$ and $[0,-1,0]$). Peaking ramps, namely $[0,1,-1]$ and $[1,-1,0]$, account for 15% and 14% of the data set, respectively. The representation of other SOM groups are all less than 10%.

Bootstrapping tests failed to identify statistically significant interactions between extended ramp shapes and ramp drivers in most cases, with the exception of plateauing ramp-ups (SOM7; $[0,1,0]$). Plateauing ramp-ups are found to be less frequent during post-frontal and non-frontal precipitation events while being more common under high turbulence and stable ABL conditions. Post-frontal and precipitation processes are indeed expected to exhibit continuously oscillating features rather than a steady increase between two power generation levels, and the opposite is true for ramps occurring under stable conditions. Overall, these findings mostly invalidate the hypothesis that different ramp drivers lead to different ramp shapes. Specific results from the bootstrap test and associated p-values are provided in Table A1 and A2 in the Appendix Section.

### 3.4 Case Studies

In this section, we present several characteristic events with further details on the association between environmental conditions and wind farm generation. The case studies are for illustrative purposes and put into perspective the ramp driver classes introduced earlier.

– **Case Study 1: High winds during ramp on 9 October 2016.** Strong winds were recorded throughout the day, with maximum wind speed exceeding 28 ms$^{-1}$. The automatic ramp categorisation scheme detected two ramp-downs at 12:26 and 13:47, directly followed by a ramp-up at 14:42. Environmental conditions centred on the ramp at 13:47 are provided in Figure 12. It is evident from Figure 12 (a) that wind power largely exceeded generated power and its variations are not accompanied by analogous changes in generated power throughout the day. This is explained by the fact that groups of turbines periodically initiated shutdown/restart procedure to prevent structural load or turbine damage. In addition, rapid changes in wind direction observed during the day have caused misalignment between the wind direction and the axis of the turbine rotor blades, hence diminishing the wind power harvested.

– **Case Study 2: Passage of a cold front on 7 December 2017.** A typical example of a ramp associated with a cold front is presented in Figure 13a, during which the power generation increased by 83 MW (63% rated power) in 26 minutes.



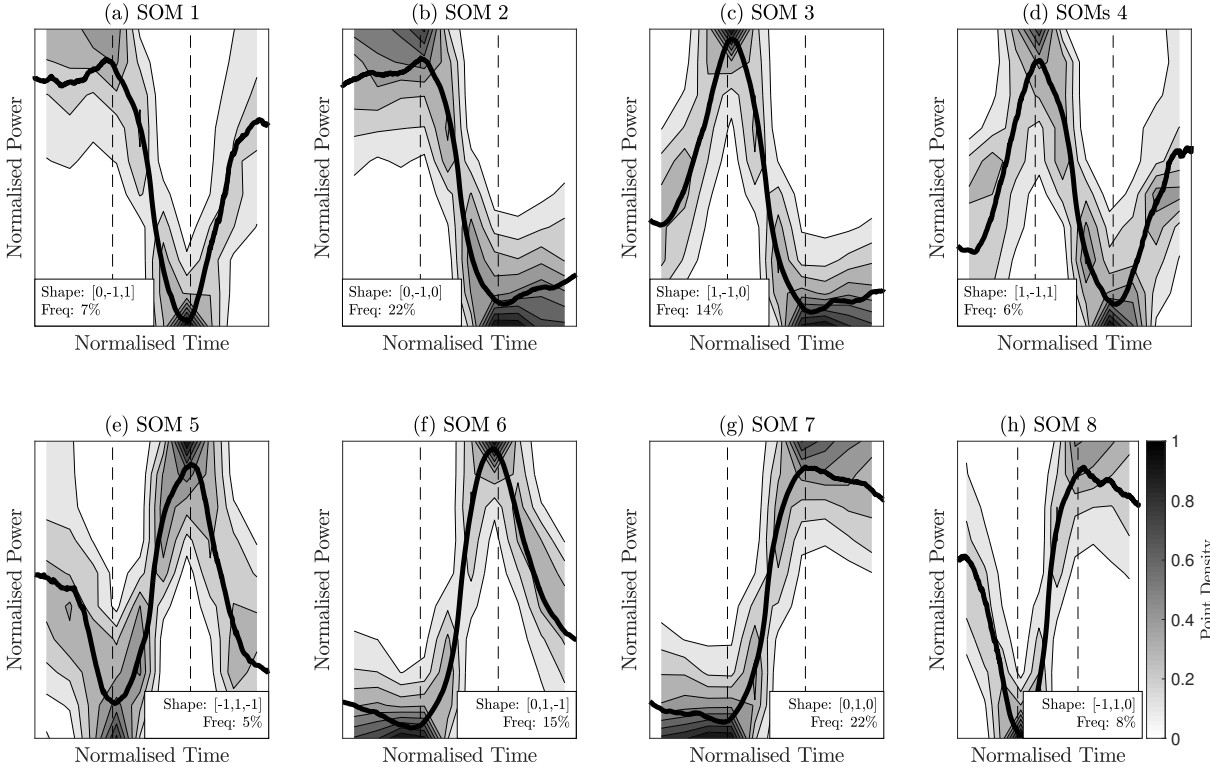

**Figure 11.** Extended ramp shapes classes resulting from SOM clustering. The thick black lines show the averaged extended ramp shape, and the grayscale contours display the point density for each SOM groups. The dashed lines delineate periods equal to one rise time.

Figure 13a shows typical cold front features such as a shift in wind from the northwest to the southwest quadrant together with decreasing temperature, increasing air density and precipitation. Analysis of the MSLP chart shortly before ramp (Figure 13b) validates the cold front classification by the automatic ramp categorisation scheme.

– **Case Study 3: Passage of a storm on 14 December 2018.** Severe thunderstorm activity was recorded throughout the day, leading to flash flooding in the region (Press, 2018). Figure 14 shows a mid-afternoon increase in relative humidity coupled with a drop in temperature and precipitation, characteristic of a passing storm cell. The increased wind speeds, likely associated with the thunderstorm downdraft, resulted in a 59 MW amplitude ramp-up centred at 16:00 (Figure 14). The relaxation in wind speeds following the passage of the storm induced a downward ramp at 16:43, with an amplitude

of 43 MW. The classification algorithm adequately categorised both ramps as non-frontal precipitation events.

    – **Case Study 4: Other - cold air outbreak with cellular convection on 25 September 2017.** As discussed in Section 2.4, not all meteorological phenomena are captured through the ramp classification scheme. An example categorised as 'other' took place at 14:34 on 25-Sep-2017; a day characterised by constant power generation fluctuations (Figure 15a). Analysis of satellite imagery around the timing of the ramp (Figure 15b) suggests the variability observed throughout

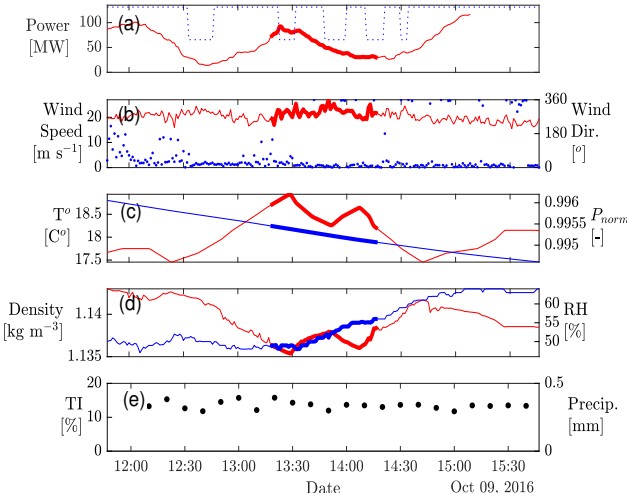

**Figure 12.** Time series of environmental variables encompassing ramp on 09-Oct-2016 at 13:47:00 [UTC+10]. (a) Total wind farm power generation (red line) and mean wind power (mean wind speed from the two onsite meteorological towers converted to wind power based on the turbine's power curve and the number of turbines actively generating; dashed blue line), (b) mean wind speed (red line) and wind direction (blue dots), (c) temperature and 30 min normalised pressure (red and blue line, respectively) (d) air density and relative humidity (red and blue line, respectively) and (e) mean horizontal turbulence intensity over a 10 min window (black dots) and precipitation recorded at the weather station neighbouring the site (Sheoaks; blue bars). Thickened red lines correspond to the extent of the ramp scale.

the day is attributable to a cold air outbreak. Such meteorological phenomena are often associated with sustained cellular convection (Vincent et al., 2012), driving wind speed fluctuations. In particular, MSLP charts indicate a low-pressure system with an embedded front passing over the site at around 0100 AEST on the same day and as such, the ramp should be considered as a particular case of post-frontal convection. While the automatic front identification scheme accurately detected the frontal passage, the ramp was not associated with post-frontal conditions because it occurred more than 12 hours after the cold front.

## 4 Conclusions

Sudden wind power variations and associated underlying processes need to be accurately characterised to enhance ramp forecast accuracy and hence reduce grid instability. Although the influence of more common (i.e. less extreme) wind power ramps are evident, the current body of literature on ramp characterisation focuses mostly on the largest ramps owing to a lack of an automated classification methodology. This paper bridges this knowledge gap by assessing power variations with a temporal coverage exceeding 5%. In this study, we introduced a robust method to characterise intra-hourly wind power ramps at the wind





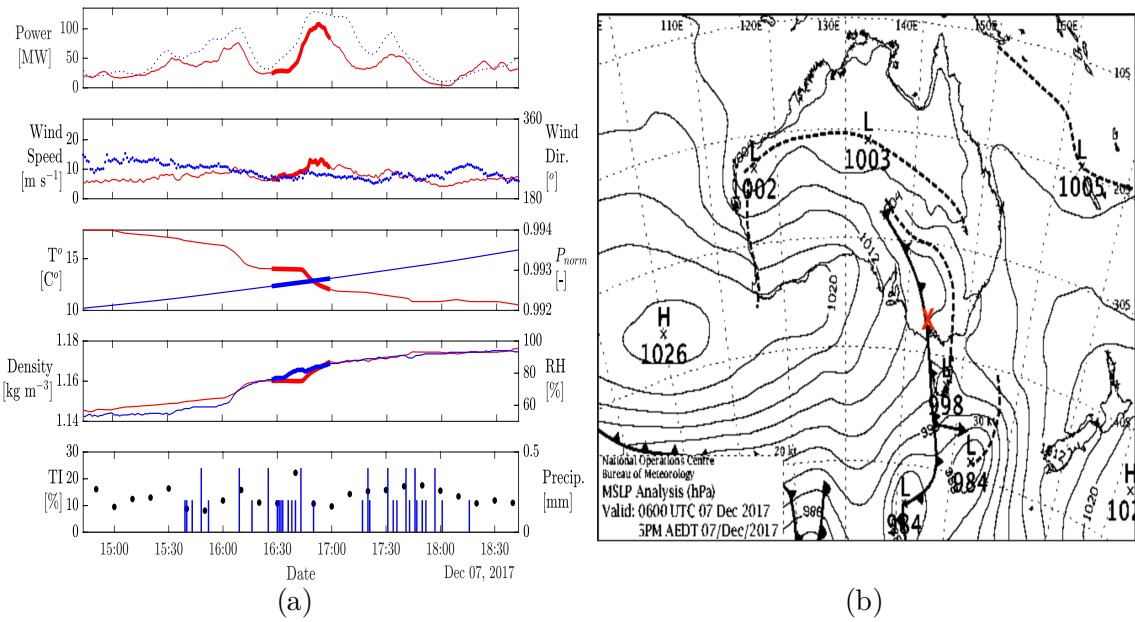

**Figure 13.** (a) time series of environmental variables encompassing ramp on 07/12/2017 at 16:43:00 [UTC+10] (b) MSLP Chart on 07/12/2017, 16:00:00 [UTC+10] (Source: Australian Bureau of Meteorology). The properties of the plots on (a) are analogous to Figure 12. The red cross on (b) indicates approximate location of the site.

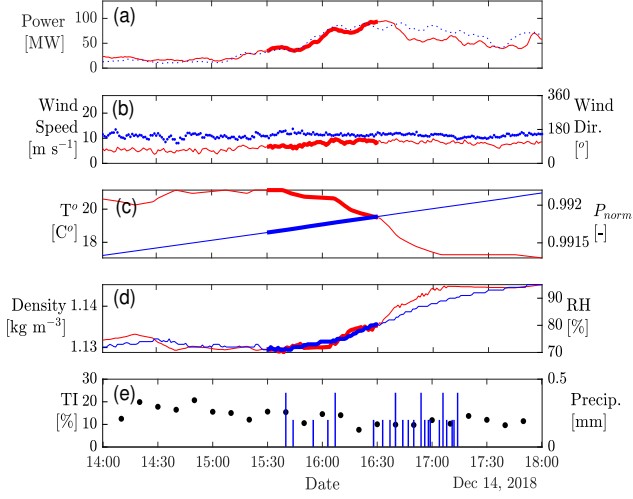

**Figure 14.** Time series of environmental variables encompassing ramp on 14-Dec-2018 at 16:00:00 [UTC+10]. The properties of the plots are analogous to Figure 12.





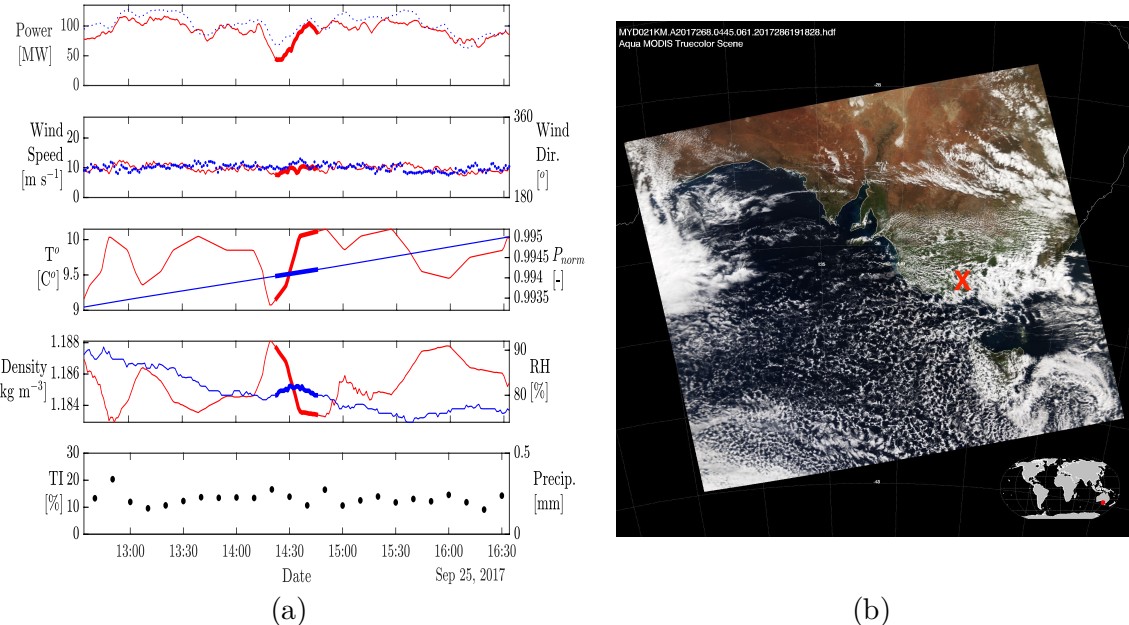

(a)                            (b)

**Figure 15.** (a) time series of environmental variables encompassing ramp on 25-Sep-2017 at 14:34:00 [UTC+10] (b) satellite imagery on 25-Sep-2017, 14:45:00 [UTC+10] (Source: https://modis-images.gsfc.nasa.gov/). The properties of the plots on (a) are analogous to Figure 12. The red cross on (b) indicates approximate location of the site.

farm scale. We then explored the underlying causes of ramps by means an automatic ramp categorisation scheme. Finally, we investigated the shape of the fluctuations surrounding ramps to improve ramp modelling.

The results are significant in three respects. First and foremost, we show how simple statistics can provide valuable insights
on the complex mechanisms shaping ramp event dynamics. Second, although the behaviour of the power fluctuations before and after a ramp can vary greatly, some ramping behaviours are more frequent than others. For instance, power fluctuations tend to plateau before and after the ramp in 44% of the cases. Such considerations need to be accounted for when modelling ramps. Third, the study showed that cold fronts and post-frontal activity accounted for most of the ramps investigated (46%), followed by precipitation events (29%). Implications in terms of forecastability are significant. As passages of cold fronts are often
predictable several days in advance using numerical weather prediction models, albeit with timing errors, these can be used to warn operators in the control room of days with high chances of ramp occurrence. Likewise, wind farm operators can expect more wind power variability within 12 hours of the passage of a front (post-frontal conditions). Similarly, precipitation events can be challenging to predict accurately more than a couple of hours in advance, particularly where stochastic convective-scale processes are present.

The results presented here indicate the potential for real-time, upstream ramp detection using remote sensing and in situ observations. Finally, we note that accurate modelling and prediction of wind power ramps are also beneficial to other areas of research, such as aviation safety and building design.



*Data availability.* MSLP charts and satellite imagery data presented in this study can be accessed online at http://www.bom.gov.au/australia/charts/archive/ and https://modis-images.gsfc.nasa.gov/, respectively. Wind farm power generation data are confidential and therefore not
publicly available. Environmental data presented in this study are available from the Bureau of Meteorology.

*Author contributions.* All authors contributed to the design and implementation of the research and the analysis of the results. MP wrote the manuscript with input from CV, GS and JM.

*Competing interests.* The authors declare that they have no conflicts of interest.

*Acknowledgements.* This study is partly funded by the Australian Renewable Energy Agency (ARENA) in the context of the Market Par-
ticipant 5-Minutes forecast (MP5F) initiative undertaken by the Australian Renewable Energy Agency (ARENA) and the Australian Energy Market Operator (AEMO). We would like to thank Meridian Energy Australia for providing the data presented in this study.





## Appendix A: Bootstrapping Method

In this paper, we use a bootstrapping approach to assess whether specific ramp drivers have characteristic ramp shapes. Bootstrapping is a common statistical test used to evaluate the sampling distribution of a variable based on random sampling. The
population is sampled a number of times equal to the number of samples (there are 1183 ramps in the data set), according to weights given by the probability distribution assuming no relationships between the ramp shapes and drivers. This random sampling with replacement is repeated 1000 times. The observed ensemble frequencies falling outside of the 95% confidence interval of their bootstrapped distributions indicate statistically significant differences. Results from the bootstrap test and associated p-values are provided in Table A1 and A2, respectively.

**Table A1.** Interactions between ramp driver and associated shape class - results from the bootstrap test in which -1 represents observations lesser than the 95% confidence interval lower-bound from the bootstrapped distribution, 1 represents observations higher than the 95% confidence interval upper-bound and 0 denotes no statistically significant differences.

|                             | SOM1 | SOM2 | SOM3 | SOM4 | SOM5 | SOM6 | SOM7 | SOM8 |
|-----------------------------|------|------|------|------|------|------|------|------|
| Cold Fronts                 | 0    | 0    | 0    | 0    | 0    | 0    | 0    | 0    |
| Post Frontal Activity       | 0    | 0    | 0    | 0    | 0    | 0    | -1   | 0    |
| Precipitation (non-frontal) | 0    | 0    | 0    | 0    | 0    | 0    | -1   | 0    |
| High Turbulence             | 0    | 0    | 0    | 0    | 0    | 0    | 1    | 0    |
| Stable ABL Effects          | 0    | 0    | 0    | 0    | 0    | 0    | 1    | 0    |
| Shutdown Cases              | 0    | 0    | 0    | 0    | 0    | 0    | 0    | 0    |

**Table A2.** Bootstrap test p-value.

|                             | SOM1  | SOM2  | SOM3 | SOM4 | SOM5 | SOM6 | SOM7   | SOM8 |
|-----------------------------|-------|-------|------|------|------|------|--------|------|
| Cold Fronts                 | 0.65  | 0.48  | 0.48 | 0.82 | 0.61 | 0.58 | 0.079  | 0.29 |
| Post Frontal Activity       | 0.17  | 0.70  | 0.16 | 0.78 | 0.74 | 0.37 | 0.033  | 0.71 |
| Precipitation (non-frontal) | 0.18  | 0.49  | 0.80 | 0.13 | 0.45 | 0.29 | 0.034  | 0.94 |
| High Turbulence             | 0.070 | 0.058 | 0.26 | 0.86 | 0.49 | 0.65 | 0.0015 | 0.16 |
| Stable ABL Effects          | 0.32  | 0.60  | 0.71 | 0.25 | 0.40 | 0.29 | 0.0075 | 0.47 |
| Shutdown Cases              | 0.50  | 0.73  | 0.32 | 0.38 | 0.55 | 0.97 | 0.69   | 0.45 |



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
