# Peer review of "Characterisation of Intra-hourly Wind Power Ramps at the Wind Farm Scale and Associated Processes"

_Wind Energy Science, 2020_

## Referee Comment (RC1) · Anonymous Referee #1 · 19 Jul 2020

In this paper, the authors present an automated characterization of wind power ramps in wind farms using data from the Australian Mount Mercer wind farm. The methodology is based on a wavelet transform through which the amplitude and duration of the ramps are detected. The authors set up a decision tree based on which the ramps are categorized into events related to cold fronts, moist convection, turbulence, etc. As a result, the paper offers a comprehensive analysis of more than a thousand ramp events spanning measurements of almost two and a half years. The paper is very well written and appears technically sound. The results promise real-world applicability in wind farm operations. Therefore, I am happy to recommend the paper for publication in Wind Energy Science in its current form.

[Figure]

Still, I would like to ask the authors to address the following points:

- Why do the authors refer to a dot product in eq.(1)? Do they mean the inner product of $p_t$ and $\Psi(t)$ in some function space? Or would it be enough to call it a product?

- Eq.(1) is missing a "dt" in the integration. I would also leave out the "\cdots".

- p5: "'Haar' Wavelet", "Wavelet" should not be capitalized.

- Please check the normalization of the PDF presented in Fig. 4.

- Instead of the scatter plot presented in Fig. 6, would it be possible to compute the joint PDF, e.g. using kernel density estimation?

---

## Referee Comment (RC2) · 13 Aug 2020

First I would like to apologise for the delay of my review.

This paper deals with a very interesting topic, namely automatic characterisation of wind power ramps at a wind farm site. The analysis consists of many well-performed steps and ambitious tasks. The paper is well structured, the introduction is well written and formulated, and the methodology section is very clear.

[Figure]

There are however some assumptions in the paper that lead to some of the final conclusions being rather unreliable. I believe that the main issue is that the available measurements at the wind farm site are quite limited and cannot accurately represent the parameters that the ramp classification is based on.

Major issues:

1. Using TI as proxy for atmospheric stability. The papers that are referenced: Wharton and Lundquist (2012) and Hansen et al. (2012) investigate the influence of stability on TI. It can be seen in these studies that the spread of TI in each stability category is simply too high to give a reliable measure of stability on a 10-minute basis. As can be seen from table 1 in the current paper, a TI of 7% would represent convective conditions according to Hansen et al. but stable conditions according to Wharton and Lundquist! In a more recent paper, some of the same authors conclude: "Analysis indicates that while TI is generally lower in the stable class this assumption cannot be uniformly applied due to the wide range of TI in both stable and unstable classes " (Barthelmie et al., The role of atmospheric stability/turbulence on wakes at the Egmond aan Zee offshore wind farm, 2015) Further, there is no filtering of wake measurements. When the wind direction is in the range approx 100-190°, the mast is placed in the wind farm wake with high turbulence intensities. The wake flow is not representative of the general weather conditions. This will lead to falsely identified high turbulence conditions. Finally, there is a ramp effect. When the wind speed increases, e.g. during a frontal passage, the 10-minute TI will be relatively high, unless the wind speed measurements are detrended. Here there is no mention of detrending, so these conditions will also be falsely identified as high turbulence conditions.

2. The classification scheme, or ramp drivers, include categories that cannot cause
ramps on a wind farm scale. These are high turbulence and precipitation. While ramps can definitely be associated with both precipitation and high turbulence, these conditions are not the causality of wind farm ramp events.

Due to the non-linearity of the wind turbine power curve, high TI (disregarding shear and veer) will have the following effect: at high wind speeds (near rated wind speed) the wind turbine power curve is reduced. Conversely, at low wind speeds (near cut-in wind speed) the power curve will increase. So, high TI will have to be conditioned on wind speed and have a sudden change to result in a wind power ramp.

Also, due to the limited scale of turbulent fluctuations (microscale range) and their inherent stochastic nature, turbulent fluctuation cannot cause an increase in the power output of all the wind turbines in the farm simultaneously.

If the main purpose of the ramp classification is to improve ramp forecasts, then it is vital that the categories are real ramp drivers. It is e.g. not possible to forecast rain and then expect a ramp at the wind farm site.

Minor issues:

1. Using precipitation data 25 km away from the site can be problematic in the summertime, with frequent convective conditions and very local rain showers. This may lead to falsely detected rain events at the wind farm site.

2. The data (temperature, pressure, density and relative humidity) shown in Figures 12-15 has not been mentioned anywhere in the text. Are these measurements from the meteorological station 25 km away?

Based on the major issues I suggest the following changes of the analysis:
Because the data on the wind farm site is quite limited, the analysis could be simplified. Focus on the ramp identification algorithm and the frontal passage detection. Include only three or four characterisation groups. These could be: Frontal passage, post

frontal activity, mesoscale fluctuations near rated wind speed, and other. This would be a first step towards a more detailed ramp classification scheme, but would be very valuable all the same. In order to define a more detailed ramp classification scheme, a more extensive data set at the wind farm site is needed.

The current paper also opens up for other interesting research questions such as: How often does a frontal passage result in a wind power ramp at the wind farm? Will all frontal passages lead to wind power ramps?

---

## Author Comment (AC1) · 15 Sep 2020

Dear reviewers,

First and foremost, I would like to express our sincere gratitude on behalf of all the co-authors for your constructive feedback and insightful suggestions. We also would like to thank the reviewers for their careful reading of the paper. All comments provided were given full consideration and led to revisions in the manuscript.

We believe the reviewers' suggestions have been very helpful in improving the article quality. Please find our responses in the document below. Please also note the

attached revised paper, including all changes marked up in red.

Please also note the supplement to this comment:
https://wes.copernicus.org/preprints/wes-2020-81/wes-2020-81-AC1-supplement.pdf

**Supplement:**

Dear reviewers,

First and foremost, I would like to express our sincere gratitude on behalf of all the co-authors for your constructive feedback and insightful suggestions. We also would like to thank the reviewers for their careful reading of the paper. All comments provided were given full consideration and led to revisions in the manuscript.

We believe the reviewers' suggestions have been very helpful in improving the article quality. Please find our responses in the document below. Please also note the attached revised paper, including all changes marked up in red.

**REVIEWER COMMENT 1 – Anonymous Referee**

In this paper, the authors present an automated characterization of wind power ramps in wind farms using data from the Australian Mount Mercer wind farm. The methodology is based on a wavelet transform through which the amplitude and duration of the ramps are detected. The authors set up a decision tree based on which the ramps are categorized into events related to cold fronts, moist convection, turbulence, etc. As a result, the paper offers a comprehensive analysis of more than a thousand ramp events spanning measurements of almost two and a half years. The paper is very well written and appears technically sound. The results promise real-world applicability in wind farm operations. Therefore, I am happy to recommend the paper for publication in Wind Energy Science in its current form.

Still, I would like to ask the authors to address the following points:
1. Why do the authors refer to a dot product in eq.(1)? Do they mean the inner product of $p_t$ and $n\Psi(t)$ in some function space? Or would it be enough to call it a product?
   We concede the terminology 'product' as opposed to 'dot product' is better suited. The text has been updated accordingly.
2. Eq.(1) is missing a "dt" in the integration. I would also leave out the "ncdots".
   That is right. The equation has been changed accordingly.
3. p5: "'Haar' Wavelet", "Wavelet" should not be capitalized.
   This has now been addressed in the text.
4. Please check the normalization of the PDF presented in Fig. 4.
   The PDF normalisation on Fig. 4 has been fixed. Thank you for pointing this out.
5. Instead of the scatter plot presented in Fig. 6, would it be possible to compute the joint PDF, e.g., using kernel density estimation?
   Good point. The scatter plot has been changed to a joint PDF distribution following the Reviewer's comments. In order to conserve insights on ramp's characteristics depending on their gradient and also avoid redundancies within the plot, the univariate PDF distributions were further revised to display the marginal distributions of upwards and downwards ramps.

This paper deals with a very interesting topic, namely automatic characterisation of wind power ramps at a wind farm site. The analysis consists of many well-performed steps and ambitious tasks. The paper is well structured, the introduction is well written and formulated, and the methodology section is very clear.

There are however some assumptions in the paper that lead to some of the final conclusions being rather unreliable. I believe that the main issue is that the available measurements at the wind farm site are quite limited and cannot accurately represent the parameters that the ramp classification is based on.

Major issues:
1. Using TI as proxy for atmospheric stability. The papers that are referenced: Wharton and Lundquist (2012) and Hansen et al. (2012) investigate the influence of stability on TI. It can be seen in these studies that the spread of TI in each stability category is simply too high to give a reliable measure of stability on a 10-minute basis. As can be seen from table 1 in the current paper, a TI of 7% would represent convective conditions according to Hansen et al. but stable conditions according to Wharton and Lundquist!
In a more recent paper, some of the same authors conclude: "Analysis indicates that while TI is generally lower in the stable class this assumption cannot be uniformly applied due to the wide range of TI in both stable and unstable classes" (Barthelmie et al., The role of atmospheric stability/turbulence on wakes at the Egmond aan Zee offshore wind farm, 2015).

Thank you for drawing attention to this important matter. We recognise there is considerable uncertainty regarding the intricate relationships between TI and stability conditions and better understanding these relationships is in itself a very interesting research topic.

It is important to note that TI thresholds for stability classification are not necessarily comparable between studies. As stated in Wharton and Lundquist (2012) 'because turbulence intensity is a relative quantity, TI thresholds appear to be very sensitive to the type of instrument and methodology used' and as a result 'the stability threshold criteria have been modified slightly according to the range of atmospheric conditions and terrain observed at this wind farm'. The 9% and 14% TI thresholds used in our study were established with these considerations in mind based on site-specific TI observations (cf. Figure 4). Unfortunately, we did not have sufficient data to calculate stability at the site in the historical period.

That being said, we understand and agree that TI measurements tend to vary substantially within each atmospheric stability class. As such, results from TI-based stability categorisation should be interpreted with extreme care. Considering the legitimate controversial aspect of atmospheric stability classification based on absolute TI thresholds, we discarded the two associated drivers from the classification scheme, namely 'high-turbulence' and 'stable boundary layer effect'. We also removed the review on TI thresholds used in the literature from the manuscript (former Table 1). Finally, we also added a new category based on TI variations, as discussed further below.

Further, there is no filtering of wake measurements. When the wind direction is in the range approx 100-190_, the mast is placed in the wind farm wake with high turbulence intensities. The wake flow is not representative of the general weather conditions. This will lead to falsely identified high turbulence conditions. Finally, there is a ramp effect. When the wind speed increases, e.g. during a frontal passage, the 10-minute TI will be relatively high, unless the wind speed measurements are detrended. Here there is no mention of detrending, so these conditions will also be falsely identified as high turbulence conditions.

We acknowledge detrending and wake filtering are two critical processing steps for TI calculation that were overlooked in the study. We have now included these in the TI calculation and updated the manuscript's methodology and result section accordingly. Figure 4, 12, 13(a), 14 and 15(a) were also updated to include revised TI estimations. For wake filtering, we used a conservative valid sector range comprised between 290° and 110° for the northwest mast (MM1) and between 79° and 259° for the southeast mast (MM2). Wind speed measurements were further linearly detrended for each 10 min window. We thank the reviewer for pointing this out and the changes made undoubtedly make the manuscript better.

2. The classification scheme, or ramp drivers, include categories that cannot cause ramps on a wind farm scale. These are high turbulence and precipitation. While ramps can definitely be associated with both precipitation and high turbulence, these conditions are not the causality of wind farm ramp events. Due to the non-linearity of the wind turbine power curve, high TI (disregarding shear and veer) will have the following effect: at high wind speeds (near rated wind speed) the wind turbine power curve is reduced. Conversely, at low wind speeds (near cut-in wind speed) the power curve will increase. So, high TI will have to be conditioned on wind speed and have a sudden change to result in a wind power ramp. Also, due to the limited scale of turbulent fluctuations (microscale range) and their inherent stochastic nature, turbulent fluctuation cannot cause an increase in the power output of all the wind turbines in the farm simultaneously. If the main purpose of the ramp classification is to improve ramp forecasts, then it is vital that the categories are real ramp drivers. It is e.g. not possible to forecast rain and then expect a ramp at the wind farm site.

We agree that highly turbulent conditions themselves are not a critical factor for causing ramps, rather circumstances under which they occur. We also agree that fast wind speed variations tend to be smoothed out at the wind farm scale due to averaging of the fluctuations along the rotor sweep, combined with the spatial aggregation of wind power over the wind farm area. Consequently, micro-scale variations do not result in wind power ramps at the wind farm scale.

To put more emphasis on the causal relationship between ramps and their drivers, we decided to include another ramp driver class called 'Large change in Turbulence Intensity'. We believe this new category provides realistic insights on changes within the atmospheric boundary layer structure while avoid drawing potentially misleading conclusions on atmospheric stability classes. We suggest two ways in which a change in vertical turbulent mixing can induce wind power ramps. Firstly, ramp-up events can occur in the instance of a high wind speed layer situated above hub height in conjunction with the erosion of a stable boundary layer. Conversely, rapid radiative cooling of the surface layers can result in a substantial increase of thermal stability hence reducing vertical momentum flux and wind power harvested at hub height. Although to a lesser extent, decrease in turbulence intensity level may also influence the potential development of wakes, low-level jets, and gravity waves, although the diagnosis of these effects is outside the scope of this study. We have added these explanations along with further details on the criterion to the main manuscript.

On the other hand, we disagree with the statement that precipitation cannot induce ramps. Showers and thunderstorms are paired with convective processes in which downdrafts play a significant role and could result in synchronous wind speed change at the wind farm scale. Further to this, the outflow from a thunderstorm can also cause gustiness, sometimes organised as a 'gust front', which in turn induces wind power ramps. For example, such events are discussed in 'Potter and Hernandez, Downdraft outflows: climatological potential to influence fire behaviour, 2017', 'Nguyen et al, Wind turbine loads during simulated thunderstorm microbursts, 2011' and 'Han et al, Fine gust front structure observed by coherent Doppler lidar at Lanzhou Airport, 2020'. Further to this point; using met masts at wind-turbine hub heights, 'Vincent et al, Wind fluctuation over the North Sea, 2011' found that severe wind speed fluctuations were found in the vicinity of precipitation. Nonetheless, we acknowledge the

manuscript lacked clear explanations on the dynamics of precipitation-related ramps and these have now been added to the text.

Minor issues:

1. Using precipitation data 25 km away from the site can be problematic in the summertime, with frequent convective conditions and very local rain showers. This may lead to falsely detected rain events at the wind farm site.

   We agree that using offsite precipitation data might lead to classification bias as rain events can occur on short spatial and temporal scales. Using offsite data to characterise onsite conditions is, to some extent, ill-founded. We recognise such limitation was not clearly stated in the paper. This is now clearly acknowledged in the methodology and result section of the revised manuscript.

   Considering the shortcomings associated with the use of offsite precipitation data, we modified the condition for precipitation-related ramps in two ways. First, we extended the time window over which precipitation can be detected from one to two hours, particularly from one hour before to one hour after the ramp's timing. This was done in an effort to effectively capture sustained precipitation systems moving towards the southeast (the Sheoaks weather station is located 25 km to the southeast of the site). Second, we added a requirement on the accumulated rain depth within the searching window to ignore light rain and emphasize on larger rainfall systems with a prolonged life cycle. In short, a ramp is associated with non-frontal precipitation when at least 1 mm of cumulative rainfall is recorded within a two-hour window centred on the ramp. The 1 mm threshold was chosen in light of the Australian Bureau of Meteorology's definition of a 'wet day', which is a day with precipitation equal or greater than 1 mm (Haylock and Nicholls. Trends in extreme rainfall indices for an updated high-quality data set for Australia, 2000).

   While we understand cumulative rainfall and precipitation length scales are not necessarily correlated, various studies suggest a strong relationship between cumulated precipitation and lifetime of the rain convective cell. This is discussed for example in 'Peleg and Morin, Convective rain cells: Radar-derived spatiotemporal characteristics and synoptic patterns over the eastern Mediterranean, 2012'. Finally, we would like to point out that neighbouring rain systems can, in some cases, result in ramps even though there was no rain recorded at the wind farm site through the ramp-inducing processes discussed above (e.g. cellular convection, gust fronts and microbursts). Such situations are discussed in 'Fournier and Haerter, Tracking the Gust Fronts of Convective Cold Pools, 2019' for instance. For these reasons, we believe the newly introduced criterion is well suited to capture ramps most likely associated with non-frontal rain events, despite the use of offsite precipitation data.

2. The data (temperature, pressure, density and relative humidity) shown in Figures 12-15 has not been mentioned anywhere in the text. Are these measurements from the meteorological station 25 km away?

   Other than the precipitation data, all the measurements originate from onsite met masts. The methodology section of the paper has been updated to make this point clear.

Based on the major issues I suggest the following changes of the analysis: Because the data on the wind farm site is quite limited, the analysis could be simplified. Focus on the ramp identification algorithm and the frontal passage detection. Include only three or four characterisation groups. These could be: Frontal passage, post fluctuations near rated wind speed, and other. This would be a first step towards a more detailed ramp classification scheme, but would be very valuable all the same. In order to define a more detailed ramp classification scheme, a more extensive data set at the wind farm site is needed.

Thank you for this recommendation. We appreciate fewer ramp driver groupings could be seen as a way to consolidate the outcomes of the study. Notwithstanding, we strongly believe the revised categories provide reliable insights into the main ramp drivers and reflect the real conditions at the site as closely as possible. Our view is that including only three ramp types would lead to losing one of the paper's main points, i.e. to better-understand circumstances under which ramps occur. In this sense, the goal of the paper is to explore associations between ramps and their drivers for use in forecasts, rather than focusing exclusively on causal relationships. We are hopeful that our improvements to the TI calculation method, our updated ramp driver classes, and the use of onsite measurement for temperature, pressure and relative humidity satisfactorily respond to your statement on the need for a more extensive dataset to substantiate the ramp classes presented in the study.

Lastly, we would like to point out the paper's goal is not only to categorise wind power ramps according to their most likely driver but also to provide insights on other important characteristics such as ramps' severity, seasonality, diurnal pattern, and shape as well as presenting some typical real-world examples.

The current paper also opens up for other interesting research questions such as: How often does a frontal passage result in a wind power ramp at the wind farm? Will all frontal passages lead to wind power ramps?

These research questions are indeed very interesting and would require further investigations. We have now acknowledged this as a consideration for future study. Thank you for the suggestion.

[revised manuscript text omitted]

---

## Author Response (AR2)

Dear reviewer,

Please find our final responses in the document below. Please also note the attached revised paper, including all changes marked up in red.

Sincerely,

Mathieu Pichault.

**REVIEWER COMMENT #2 – Asta Hannesdóttir**

Dear authors,

I believe the manuscript is much improved by the review process and all issues have been dealt with in a satisfactory manner.

I only have a few minor comments and questions:
1) Figure 2: There is a Lidar in the figure that isn't relevant for this paper. I would suggest removing it from the figure. Also, the image is from Google Earth and there might be a copyright issue. Can you check with the editor?
The figure has been updated accordingly, and a copyright statement has been added to the figure caption.

2) p. 8, line 194: I don't quite understand this sentence: "… the most likely timing for each front was selected with a cumulative sum over a 6-hour window ." Cumulative sum of what? Could this be explained better?
The text has been updated to include a more precise explanation.

3) Figure 12: The figure could be made larger, so it becomes easier to read the axis labels and titles. Also, the sudden jumps in the directional data (from ≈0° to ≈180°) in plot b) might suggest that the angles have been averaged with an arithmetic mean, which does not make sense for for angles. Could you check if you are using a proper mean of angles for the directional data?
The size of Figure 12 has been increased. As far as angle averaging is concerned, I can confirm appropriate angular averaging was performed throughout the study.

Best regards,
Ásta

[revised manuscript text omitted]